# Relationship between cricket participation, health and well-being: scoping review protocol

Garrett Scott Bullock  ,[1] Nirmala K Panagodage-Perera,[1] Andrew Murray,[2] Nigel K Arden,[1] Stephanie R Filbay  [1]

¹Arthritis Research UK Centre for Sport, Exercise and Osteoarthritis, Nuffield Department of Orthopaedics, Rheumatology, and Musculoskeletal Sciences, University of Oxford, Oxford, UK
²Physical Activity for Health Research Centre, University of Edinburgh, Edinburgh, UK

**Correspondence to**
Dr Garrett Scott Bullock;
garrett.bullock@wolfson.ox.ac.uk

## ABSTRACT

**Introduction** Cricket is a popular sport played by 2.5 billion people of all ages and abilities. However, cricket participation is decreasing in the UK, despite an increased focus of governments on increasing sport participation to enhance public health. Understanding the health benefits and mitigating the health risks of cricket participation may help cricket organisations promote cricket participation while optimising the long-term health of cricket participants. Currently, there is no literature review on the relationship between cricket participation, health and well-being; thus, this relationship remains unclear. Therefore, the aims of this scoping review were (1) to investigate the relationship between cricket participation, health and well-being and (ii) to identify the research gaps related to cricket, health and well-being.

**Methods and analysis** Due to the broad nature of our research question and the large number of health outcomes assessed within the cricket literature and to facilitate identification of research gaps, a scoping review methodology was used. The methodology of this paper was informed by previous scoping review protocols and best practice methodological frameworks. MEDLINE, CINAHL, Embase, Scopus, PsycINFO, SPORTDiscus, Cochrane Library, EBSCO, Web of Science and PEDro and grey literature sources (Google Scholar, ClinicalTrials.gov, ISRCTN Registry and ProQuest) will be systematically searched. Studies that assess a construct related to health and/or well-being in current and/or former cricketers from all ages and standards of play will be eligible. Two reviewers will independently screen full texts of identified studies for eligibility and will perform data extraction. Results will be presented in tabular and graphical forms and will be reported descriptively.

**Ethics and dissemination** This research is exempt from ethics approval due to the data being available through published and public available resources. Results will be published in a peer-reviewed sports and exercise medicine journal regardless of positive or negative findings. In addition, results will be disseminated through multiple platforms, including conference presentations and social media using multimedia resources (eg, infographics, animations, videos, podcasts and blogs), to engage stakeholder groups, including cricketers, cricket coaches, sporting bodies, sports medicine professionals and policy makers. There findings will inform clinical decision making, policy changes and future research agendas.

## Strengths and limitations of this study

► Scoping reviews are a scientifically validated method to answer broad research questions and are the best methodology to provide an overview of all literature investigating the relationship between cricket participation, health and well-being.

► This scoping review will include grey literature to increase the scope and breadth of the review and screening process.

► Individual article data will be meta-aggregated to explore emergent themes, with potential to provide new insights and inform future research.

► Specific articles cannot be analysed for methodological risk of bias, decreasing the interpretability of the results.

## INTRODUCTION

Cricket is a popular sport with approximately 2.5 billion people of all ages and abilities participating.[1] Cricket is played by 1.4 million people in Australia,[2] nearly 300 000 people in the UK[3] and over 5 million people in India.[4] Further, cricket is popular among youth,[5 6] many of whom continue to play cricket into adulthood.[7] Cricket has also become increasingly popular among women, with more than 27% of all Australian cricketers being female.[2] Cricket is played with 11 individuals per team, over 5 days (test cricket), 1 day (50 overs) or over 4 hours (Twenty20). The 2011 Compendium of Physical Activities lists cricket as a sport that can provide moderate-intensity physical activity.[8] Regular physical activity is an important determinant of general health, life expectancy[9] and overall well-being.[10]

Over 31% of all adults worldwide are physically inactive, with physical inactivity levels ranging from 17% in Asia to 43% in North America.[11] To counteract inactivity, sports participation is promoted.[12] Sport participation provides opportunities to be physically active across the lifespan.[13 14] Cricket participation can improve fitness[15 16] and strength[16]

and has psychological benefits for participants,[17–19] including improved self-esteem, social connections and overall well-being.[17] Mental health and health-related quality of life (HRQoL) is higher in cricketers than in the general population.[20 21] However, cricket participation is associated with injury,[22–25] which can result in persistent joint pain and post-traumatic osteoarthritis.[26] Specifically, injury incidence has been reported to be up to 53 injuries per 10 000 athlete exposures,[27] with former cricketers reporting greater osteoarthritis compared with former rugby players.[28] Further, some cricketers experience increased levels of stress[29 30] and depression,[31 32] which can negatively impact HRQoL.[26 33 34] Thus, cricket participation may have both positive and negative impacts on health and well-being. Due to the high rate of global physical inactivity,[11] the worldwide popularity of cricket[2–4] and its viability as an outlet for physical activity across the lifespan,[13 14] information regarding the potential risks and benefits of participation in specific sports is needed to enable informed decision making for participants.

The link between sport participation and beneficial health outcomes has been synthesised in previous systematic reviews for golf, cycling, and sport and dance.[35–37] These studies found that participation in these activities had a positive relationship with physical health and well-being.[35–37] However, the relationship between cricket, health and well-being has not been investigated. Thus, there is a need to map the current evidence related to cricket, health and well-being and to identify key research priorities. This overview would also enable key stakeholders (including cricket participants, health professionals and sporting bodies) to make evidence-informed decisions relating to cricket participation. Specifically, these data will inform stakeholders on the health and well-being risks and benefits of cricket participation in order to make individual and organisational decisions on the viability of promoting cricket participation as a health-enhancing form of physical activity at different standards of play and for different age groups. Further, identifying the gaps in the literature will allow specific cricket-related research to be initiated to improve cricket participant health and well-being. Therefore, the aims of this scoping review were (1) to investigate the relationship between cricket participation, health and well-being at all ages and standards of play and (2) to identify research gaps in the existing literature on cricket, health and well-being.

## METHODS
### Patient and public involvement
No patients were involved in the design or planning of this study. However, findings from two qualitative studies investigating the relationship of physical activity and quality of life in former elite cricketers[38 39] highlighted a need for further research investigating the relationship between cricket participation and health. It was determined that a scoping review would provide valuable information regarding the relationship between cricket participation,

health and well-being while identifying key knowledge gaps to guide future research agendas. An international stakeholder group will be established, comprising current and former cricketers, cricket coaches, sports medicine professionals, cricket-related researchers and representatives from cricket sporting bodies. This key stakeholder group will meet virtually to discuss preliminary results and interpretation of findings, review a draft of the manuscript and provide input into the plan for dissemination of research findings.

### Study design
The purpose of a scoping review is to describe all available evidence underpinning a given research question, drawing on research from all possible sources; consequently, scoping reviews are broad in nature. This is in comparison to a systematic review that can only investigate one specific topic; as a result, a scoping review methodology was commenced.[40 41] The framework adopted for this scoping review follows existing best practice methodology.[40–44] The methodology was guided by the recommended five-stage process: identify the research question; identify relevant studies; select articles using a priori inclusion/exclusion criteria; chart data; and collate, summarise and report results.[40–42] The proposed study is planned to be commenced during September 2019 and is estimated to conclude in March 2020.

### Stage 1: identify the research question
The general research question was developed through exploration of the literature, multidisciplinary group discussions and collaboration with experts in cricket. To reflect the context, content and the population included in the review,[40 41] the following broad research question was proposed: *what is known about the relationship between cricket participation, health and well-being?*

### Stage 2: identify relevant studies
#### A preliminary search to identify keywords and index terms
A preliminary search was conducted on the major clinical and grey literature databases.[40–42] Databases included MEDLINE, Google Scholar and ProQuest Dissertations & Theses Global. Consistent with previous studies,[42] exploratory search terms were kept broad, to be as inclusive as possible. Search terms included 'cricket', 'health' and 'review'. The exploratory search found 37 articles in MEDLINE. The first 200 articles in Google Scholar were searched. Twenty-eight articles were identified as pertinent from MEDLINE and Google Scholar. No relevant articles were found in ProQuest. These 28 articles' references were then searched for further relevant articles.

The titles and abstracts of these 28 articles were then analysed for relevant search terms. The preliminary search identified a large number of irrelevant studies involving cricket insects and cadaveric or in vitro investigations; consequently, the search terms were updated to exclude articles with cadaver* or 'in situ' or 'in vitro' or 'insects' in the title and/or abstract. The final search

**Table 1** Inclusion and exclusion criteria

| Inclusion criteria | Exclusion criteria |
|---|---|
| ► Assesses a construct related to health (eg, injury, pain, physiological function, physical activity, tobacco use, alcohol use, body mass index, nutrition, diabetes and cardiovascular disease) and/or well-being (eg, mental health, depression, mood, anxiety, health-related quality of life and resilience) in current and/or former cricketers (of any age, sex or competition level). | ► Cricket performance parameters (eg, bowling speed, wins and losses, and bowling average). |
| ► Primary research studies, reviews, meta-analyses, guidelines or grey literature (including unpublished and ongoing trials, annual reports, dissertations and conference abstracts). | ► Biomechanics (force, torque, kinematics and electromyography) and joint range of motion/flexibility. |
| ► Human studies. | ► Cadaveric or in situ model studies. |
| ► Articles published in English. | ► Editorials, periodicals and letters to the editor. |

strategy was created to keep the search broad for greatest inclusion while excluding specific irrelevant studies identified through the preliminary search. A medical librarian assisted by ensuring the search syntax was appropriate for each database.

### Search strategy

Ten databases (MEDLINE, CINAHL, Embase, Scopus, PsycINFO, SPORTDiscus, Cochrane Library, EBSCO, Web of Science and PEDro) will be electronically searched. Google Scholar, ClinicalTrials.gov, ISRCTN Registry and ProQuest Dissertations & Theses Global will be searched for grey literature. The search strategy will be as follows: '*cricket** NOT (*cadaver** or *'in situ'* or *'in vitro' or animals or insects*)' *(*online supplementary appendix 1*). Articles will be tracked in EndNote X9 (Clarivate Analytics, 2018).

### Study eligibility criteria

Inclusion and exclusion criteria are listed in table 1.

### Stage 3: study selection

Titles and abstracts will be screened by the lead author (GSB) for eligibility, and full-text articles will be retrieved and screened by the same author (GSB) against the inclusion/exclusion criteria. A second author (NKP-P) will complete the same screening process on a random sample of 10% of the articles.[42] Any title and abstract screening disputes will be resolved through the consensus of the two authors. If concordance is less than 90%, the full title and abstract screening will be performed by the second author (NKP-P). Following title and abstract screening, the full text of all potentially eligible articles will be retrieved. First, we will attempt to access articles through university online library portals. The online library portals will be available through collaborating institutions in the UK, Sweden, Australia and the USA. If the article cannot be retrieved through the university online library portals, the authors will be contacted to request full text, and, if required, interlibrary loan with the assistance of a librarian will be attempted. If a full-text article cannot be retrieved following consultation

with a librarian, it will be excluded from the review.[42] If there are any discrepancies following full-text screening, a third author (SRF) will arbitrate all disputes and decide on final article inclusion.

### Stage 4: data extraction

Data extraction procedures will follow best systematic review practice guidelines.[45] Data will be extracted by the lead author (GSB) and inputted into a customised electronic database. The customised electronic database will be based on the National Institute for Health and Care Excellence evidence tables.[46] Quantitative data that will be extracted will include publication year, study type (primary, secondary or grey literature), country of origin, age group, competition level, study design, study description, surgical procedure (if applicable), analysis design and key findings. Qualitative data will be extracted through qualitative synthesis of related topics.[45] A second author (NKPP) will perform data extraction on 10% of the studies, selected at random. Any discrepancies in data extraction between reviewers will result in the second reviewer (NKP-P) extracting data from all studies. Following this, extracted data will be cross-checked for discrepancies, and any differences in data will be resolved between reviewers. Outcome data will be stratified into a priori themes of musculoskeletal health, general health and well-being.

### Stage 5: collating, summarising and reporting the results

Descriptive data and key findings will be collated and summarised for descriptive analysis, and the results will be presented numerically and thematically. Individual specific study quantitative and qualitative data and thematic data will be grouped into each a priori theme. Following the grouping of articles into each a priori theme, individual article data will be meta-aggregated to explore potential emergent themes.[45] Specifically, quantitative data will be extracted, sorted into relevant themes (eg, musculoskeletal health, mental health and physiological health) and descriptively reported.[45] Qualitative data

will be synthesised in Excel through a six-stage process. This six-stage process includes becoming familiar with the data, generating initial codes, searching for themes, reviewing themes, defining themes and writing up.[47 48] Research gaps will be explored and tabulated through a priori theme and emergent theme collation. Specific article data will be tabulated, and pertinent information will be aggregated into overall study data range for summarisation. Scoping review results will be presented in numeric and graphical representation for year of publication, geographical origin of publication and a priori themes. A flowchart will be created to visually detail the screening and review process.[43] Emergent themes will be presented in tabular format, along with a narrative description of results.

## ETHICS AND DISSEMINATION

This research is exempt from ethical approval since it is a review of previously published articles.

This scoping review of cricket health and well-being is novel and will provide an overview of associations between cricket, health and well-being. Further, key research priorities relevant to stakeholders in cricket, including policy makers and sports governing bodies, will be clarified by this work.

Results will be published in a peer-reviewed sports and exercise medicine journal, with open access to increase information dissemination, regardless of positive or negative findings of the relationship between cricket participation, health and well-being. In order to enhance knowledge translation of the findings, a multimodal approach will be used for dissemination. Findings will be presented at conferences, and multimedia resources (eg, infographics, animations, videos, podcasts and blogs) will be created to disseminate findings via various social media platforms and through media release.

## CONCLUSION

The aims and methodological study design were created in concordance with cricket stakeholders (current and former cricketers, physicians, physiotherapists and governing bodies) in order to have a greater understanding of the relationship between cricket participation, health and well-being. Scoping reviews are a scientifically validated method to answer broad research questions and summarise the knowledge gaps in this field. This scoping review will inform individuals and other stakeholders about the risks and benefits of cricket participation at all ages and standards of play. These findings may inform clinical decision making, policy changes and future research agendas.

**Contributors** GSB, NKP-P, NKA and SRF conceived the study idea. GSB, NKP-P, AM, NKA and SRF were involved in methodological design and planning. GSB, NKP-P and SRF wrote the first draft of the manuscript. GSB, NPP, AM, NKA and SRF critically revised the manuscript. GSB, NKP-P, AM, NKA and SRF approved the final version of the manuscript.

**Funding** This study was funded by Centre for Sport, Exercise and Osteoarthritis Research Versus Arthritis (grant reference 21595).

**Disclaimer** The Centre for Sport, Exercise and Osteoarthritis Research Versus Arthritis had no role in study design, data collection, data analysis and interpretation, manuscript preparation, nor in the decision to submit the paper for publication.

**Competing interests** None declared.

**Patient consent for publication** Not required.

**Provenance and peer review** Not commissioned; externally peer reviewed.

**ORCID iDs**
Garrett Scott Bullock http://orcid.org/0000-0003-0236-9015
Stephanie R Filbay http://orcid.org/0000-0002-9624-0791

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
