## [Reviewer comments · BMJ Open]

ARTICLE DETAILS

TITLE (PROVISIONAL)	The relationship between cricket participation, health, and wellbeing: A scoping review protocol
AUTHORS	Bullock, Garrett; Panagodage-Perera, Nirmala; Murray, Andrew; Arden, Nigel; Filbay, Stephanie

VERSION 1 – REVIEW

REVIEWER	Louise Mansfield Brunel University London UK
REVIEW RETURNED	28-Jun-2019

GENERAL COMMENTS	Many thanks for the opportunity to review this paper. I think reviews on sport, health and wellbeing are extremely relevant and worthwhile in the current political and research climate in which sport is being prioritised and investigated for health and wellbeing outcomes. I do have some comments meant to support the development of the paper. I think the abstract needs to be clearer. A stronger rationale about why this review is important to cricket organisations, players and the wider sport sector is needed. Is it an injury issue? What type of injuries? Who is driving the agenda? In addition, clarity from the outset on the standards being used for this scoping review are needed. I did wonder why you weren't doing a full systematic review? The abstract also does not indicate the detail of the participants in the review. It says players / former players but what level? I was not clear what was meant by a multi-modal approach. Coproduction work should be underpinned by evidenced-based theory and this should be made clear (and referenced – see example below) Daykin, N., Mansfield, L., Payne, A., Kay, T., Meads, C., D'Innocenzo, G., Burnett, A., Dolan, P., Julier, G., Longworth, L. and Tomlinson, A., 2017. What works for wellbeing in culture and sport? Report of a DELPHI process to support coproduction and establish principles and parameters of an evidence review. Perspectives in public health, 137(5), pp.281-288. I'd expect reference to stakeholder involvement at different stages of the project and for identified purposes. As it reads, the stakeholder work sounds like an add-on which is a terribly ineffective way to create any kind of impact (I am sure this is simply about style). Why is cricket not key word? You argue that scoping reviews are scientifically validated. I can't see that this is true at all but perhaps I do not understand what you actually mean? You also argue scoping reviews to be a best
--

	method and I'm not convinced of that either. A full systematic review including grey literature could be argued to be the best method. Perhaps a scoping review is a pragmatic method for a topic with emerging evidence? The claim about meta-aggregation being a strength is not supported. You really need to identify an established method of data synthesis. I mean if you find quantitative data what will you do with it and how will you synthesize it. The same goes for the qualitative data. Some clarity on data synthesis needed from the outset. There are up-to-date reviews of sport health and wellbeing missing from the background e.g. Mansfield et al (2018) on sport and dance, and Oja's work on cycling. I do think this kind of review protocol needs to reflect the current literature more specifically. Oja, P., Titze, S., Bauman, A., De Geus, B., Krenn, P., Reger-Nash, B. and Kohlberger, T., 2011. Health benefits of cycling: a systematic review. Scandinavian journal of medicine & science in sports, 21(4), pp.496-509. Mansfield, L., Kay, T., Meads, C., Grigsby-Duffy, L., Lane, J., John, A., Daykin, N., Dolan, P., Testoni, S., Julier, G. and Payne, A., 2018. Sport and dance interventions for healthy young people (15–24 years) to promote subjective well-being: a systematic review. BMJ open, 8(7), p.e020959. The background also needs to make clear what the impact of this evidence can really be for cricket players, organisations and the sport sector. I can see a cross reference to relevant methods for scoping reviews but it is rather a vague commentary in the text. It's difficult to know what standards are being followed at every stage. There is information missing such as date ranges for searches for example. Sorry if I have missed this but referring to standards and guidelines rather than papers that have employed scoping reviews will strengthen this protocol. Inclusion / exclusion criteria are probably best represented in a table. On page 9 it read 'the scoping review is novel'. Do you mean 'this scoping review of XXXX is novel'? Needs clarifying because a scoping review is not novel in and of itself. The funding seems to refer to a grant linked to Arthritis – is this health condition relevant to the rationale for the review. I do think much more needs to be said about the way in which this particular topic was developed with the identified stakeholders if it's relevant. A stronger link to the use of the evidence by stakeholders would then strengthen the case for the scoping review approach
--	--

REVIEWER	Paul Downward School of Sport, Exercise and Health Sciences, Loughborough University, UK.
REVIEW RETURNED	16-Aug-2019

GENERAL COMMENTS	Dear authors,
---------------

	There are a few minor issues that I feel that you should address in the manuscript. The most important one is to clarify the distinction between a scoping and systematic review. The former as a more preliminary and exploratory investigation (See Rebecca Armstrong, Belinda J. Hall, Jodie Doyle, Elizabeth Waters, 'Scoping the scope' of a cochrane review, Journal of Public Health, Volume 33, Issue 1, March 2011, Pages 147–150, https://doi.org/10.1093/pubmed/fdr015). However, you argue that the scoping review is the best methodology to use in your case. Why? and would not a systematic review be better. Should this work be a precursor to that? The impact of cricket on health is likely to be highly geographically specific, reflecting its historic development. Is allowance made for this both in terms of the language of research papers extracted and the context in which cricket is played in terms of the social determinants of health. More clarification is need on the links between the preliminary and actual searches discussed on p.6. The nature and role of the librarian and their access to research papers should be clarified on p.8 line 36. There are some minor issues of expression to resolve. p.2 line 19 'The Methodology...' p.2 line 38 'data being available' not 'data is available...'
--	--

REVIEWER	Ivan Cavero-Redondo Universidad de Castilla-La Mancha
REVIEW RETURNED	19-Aug-2019

GENERAL COMMENTS	I think that the topic is of little scientific interest and it also seems that it is not necessary to develop a protocol for a scoping review for this topic. I think it would have something more of interest if it were a systematic review of studies whose objective was to analyze the effect of cricket in the wellbeing of its participants or any particular health outcome.
--

VERSION 1 – AUTHOR RESPONSE

Reviewer: 1

Reviewer Name: Louise Mansfield

Institution and Country: Brunel University London UK

Please state any competing interests or state 'None declared': None

Please leave your comments for the authors below

Reviewer comment 2.1: Many thanks for the opportunity to review this paper. I think reviews on sport, health and wellbeing are extremely relevant and worthwhile in the current political and research climate in which sport is being prioritised and investigated for health and wellbeing outcomes. I do have some comments meant to support the development of the paper.

Author response 2.1: The authors would like to thank the reviewer for their contributions and suggestions to this manuscript.

Reviewer comment 2.2: I think the abstract needs to be clearer. A stronger rationale about why this review is important to cricket organisations, players and the wider sport sector is needed. Is it an injury issue? What type of injuries? Who is driving the agenda?

Author response 2.2: The authors would like to thank the reviewer for this question. Increased information on the background and greater clarity has been added to the abstract on the rationale behind why this study needs to be undertaken.

Author action 2.2: On page 2 lines 8-22, the abstract introduction has been elaborated as follows, "However, cricket participation is decreasing in the UK, despite an increased focus of governments on increasing sport participation to enhance public health. Understanding the health benefits, and mitigating the health risks of cricket participation may help cricket organisations promote cricket participation whilst optimising the long-term health of cricket participants. Currently there is no literature review on the relationship between cricket participation, health, and wellbeing; thus, this relationship remains unclear.

Reviewer comment 2.3: In addition, clarity from the outset on the standards being used for this scoping review are needed. I did wonder why you weren't doing a full systematic review?

Author response 2.3: The authors would like to thank the reviewer for this question. Scoping reviews are designed to offer a systematic approach to investigate broad questions and identify gaps in the literature. Further, scoping reviews are designed to create an understanding of the breadth of the literature and direction for future research. This is in comparison to systematic reviews and/or meta-analyses which are designed to amalgamate the literature on a focused study question, where large consideration to study design and quality is given. The aim of this scoping review is to investigate the relationship between cricket participation, health, and wellbeing at all ages and standards of play, and to identify research gaps. Due to the broad nature of these aims and the wealth of literature on this topic, a scoping review is the best systematic methodology to be employed. Within the abstract, greater clarity on the reasoning behind using a systematic scoping methodology has been added.

Author action 2.3: Page 2 lines 28-36, a statement has been added concerning the reasoning for using a scoping review as follows, "Due to the broad nature of our research question, the large number of health outcomes assessed within the cricket literature, and to facilitate identification of research gaps, a scoping review methodology was utilised."

Reviewer comment 2.4: The abstract also does not indicate the detail of the participants in the review. It says players / former players but what level?

Author response 2.4: The authors would like to thank the reviewer for bringing this to our attention. Greater clarity has been added relating to the cricket participants ages and standards-of-play.

Author action 2.4: On page 2 lines 45-47 The statement has been revised as follows, "Studies that assess a construct related to health and/or wellbeing in current and/or former cricketers from all ages and standards-of-play will be eligible."

Reviewer comment 2.5: I was not clear what was meant by a multi-modal approach. Coproduction work should be underpinned by evidenced-based theory and this should be made clear (and referenced – see example below)

Daykin, N., Mansfield, L., Payne, A., Kay, T., Meads, C., D'Innocenzo, G., Burnett, A., Dolan, P., Julier, G., Longworth, L. and Tomlinson, A., 2017. What works for wellbeing in culture and sport? Report of a DELPHI process to support coproduction and establish principles and parameters of an evidence review. *Perspectives in public health*, 137(5), pp.281-288.

I'd expect reference to stakeholder involvement at different stages of the project and for identified purposes. As it reads, the stakeholder work sounds like an add-on which is a terribly ineffective way to create any kind of impact (I am sure this is simply about style).

Author response 2.5: The authors would like to thank the reviewer for this comment. This specific scoping review protocol is based off of previous stakeholder work involving qualitative. This is in consensus with previous research (Daykin et al) in which the stakeholders held high regard to qualitative work as a basis for determining key themes in relation to wellbeing in sport. These qualitative studies involved retired cricketers focusing on their current physical activity and quality of life (please see full study references below). From these initial qualitative studies, we identified a need for further research to better understand the relationship between cricket participation, health and wellbeing. Further, informing research strategy throughout the process with stakeholders allows for improved research pertinence and application. Due to this, a stakeholder group involving former cricket players, coaches, clinicians, many of whom from multiple countries, will be involved in this scoping review. These stakeholders will be involved in assisting in interpreting the results, and helping in disseminating findings.

Filbay SR, Bishop FL, Peirce N, Jones ME, Arden NK. Physical activity in former elite cricketers and strategies for promoting physical activity after retirement from cricket: a qualitative study. *BMJ Open*. 2017 Nov 17;7(11).

Filbay SR, Bishop FL, Peirce N, Jones ME, Arden NK. Common attributes in retired professional cricketers that may enhance or hinder quality of life after retirement: a qualitative study. *BMJ Open*. 2017 Jul 26;7(7).

Concerning the multi-modal approach, this study will be disseminated through the stakeholder group through multiple outlets including presentations to policy makers and sports governing bodies, publishing this work in a sport and exercise medicine journal, multimedia publications including infographics and videos, and podcasts.

Author action 2.5: On page 6 lines 3-27, a section has been added concerning stakeholder involvement as follows, "Findings from two qualitative studies investigating the relationship of physical activity and quality of life in former elite cricketers,38 39 highlighted a need for further research investigating the relationship between cricket participation and health. It was determined that a scoping review would provide valuable information regarding the relationship between cricket participation, health and wellbeing, whilst identifying key knowledge gaps to guide future research agendas. An international stakeholder group will be established, comprising current and former cricketers, cricket coaches, sports medicine professionals, cricket-related researchers and representatives from cricket sporting bodies. This key stakeholder group will meet virtually, to discuss preliminary results and interpretation of findings, review a draft of the manuscript, and provide input into the plan for dissemination of research findings.

On page 3 lines 6-15, the statement has been further clarified as follows, "In addition, results will be disseminated through multiple platforms including conference presentations and social media using multimedia resources (e.g. infographics, animations, videos, podcasts, and blogs) to engage stakeholder groups including cricketers, cricket coaches, sporting bodies, sports medicine professionals and policy makers."

Reviewer comment 2.6: Why is cricket not key word?

Author response 2.6: Thank you for bringing this to our attention. Cricket was not added as a key word due to it being in the title.

Reviewer comment 2.7: You argue that scoping reviews are scientifically validated. I can't see that this is true at all but perhaps I do not understand what you actually mean? You also argue scoping reviews to be a best method and I'm not convinced of that either. A full systematic review including grey literature could be argued to be the best method. Perhaps a scoping review is a pragmatic method for a topic with emerging evidence?

Author response 2.7: Scoping reviews are designed to offer a systematic approach to investigate broad questions and identify gaps in the literature. Scoping reviews also incorporate grey literature, which further broadens the literature breadth. Scoping reviews, like systematic reviews, incorporate all of the searched literature, and then this literature is systematically screened (title/abstract followed by full-text). Further, PRISMA has also incorporated specific guidelines for scoping review methodological reporting, which has been submitted as supplementary material. Due to these methodological considerations, scoping reviews are a repeatable systematic approach. For more information, please refer to references Arksey et al. and Levac et al., Peters et al., and Munn et al. below. These specific scoping review methodological studies have been referenced within the methods for further methodological rigor.

Arksey H, O'Malley L. Scoping studies: towards a methodological framework. *International journal of social research methodology* 2005;8(1):19-32.

Levac D, Colquhoun H, O'Brien KK. Scoping studies: advancing the methodology. *Implementation science* 2010;5(1):69.

Peters MDJ , Godfrey CM , Khalil H , et al . Guidance for conducting systematic scoping reviews. *Int J Evid Based Healthc* 2015;13:141–6.doi:10.1097/XEB.000000000000050

Munn Z, Peters MD, Stern C, Tufanaru C, McArthur A, Aromataris E. Systematic review or scoping review? Guidance for authors when choosing between a systematic or scoping review approach. *BMC medical research methodology*. 2018 Dec;18(1):143.

Further, scoping reviews are designed to create an understanding of the literature and direction for future research. This is in comparison to systematic reviews and/or meta-analyses which are designed to amalgamate the literature on one specific question. The aim of this scoping review is to investigate the relationship between cricket participation, health, and wellbeing at all ages and standards of play, and to identify research gaps. Due to the broad nature of these aims, we believe a scoping review is the best systematic methodology to be employed.

Author action 2.7: On page 7, greater referencing rigor has been incorporated into the methods in the general study design, stage 1, and stage 2.

Reviewer comment 2.8: The claim about meta-aggregation being a strength is not supported. You really need to identify an established method of data synthesis. I mean if you find quantitative data what will you do with it and how will you synthesize it. The same goes for the qualitative data. Some clarity on data synthesis needed from the outset.

Author response 2.8: The authors would like to thank the reviewer for this comment. Munn et al.'s specific methodological strategy has been referenced to provide a more specific and repeatable data extraction and synthesis method for this study (see full reference below). For quantitative data, greater detail has been added for specific data points that will be extracted which will include publication year, study type (primary, secondary, or grey literature), country of origin, age group, competition level, study design, study description, surgical procedure (if applicable), analysis design, and key findings. For qualitative data, Munn et al.'s data extraction method, will be incorporated. This involves meta-aggregating themes derived from all qualitative studies.

In terms of data synthesis, greater clarity has been added to quantitative and qualitative data synthesis. Specifically quantitative data will be collated in tabular format. These data will be vote counted and descriptively analysed. Qualitative data will be thematically analysed based on the methodological process described by Maguire et al. (reference below). This six stage process will entail becoming familiar with the data, generating initial codes, searching for themes, reviewing

themes, defining themes, and writing up. Qualitative data will be synthesized in excel, which has been previously described in a study by Bree et al. (reference below)

Munn Z, Tufanaru C, Aromataris E. JBI's systematic reviews: data extraction and synthesis. *AJN The American Journal of Nursing* 2014;114(7):49-54.

Maguire M, Delahunt B. Doing a thematic analysis: A practical, step-by-step guide for learning and teaching scholars. *AISHE-J: The All Ireland Journal of Teaching and Learning in Higher Education*. 2017 Oct 31;9(3).

Bree RT, Gallagher G. Using Microsoft Excel to code and thematically analyse qualitative data: a simple, cost-effective approach. *AISHE-J: The All Ireland Journal of Teaching and Learning in Higher Education*. 2016 Jun 30;8(2).

Author action 2.8: On page 10 lines 10-20, greater clarity has been added to the data extraction methodology as follows, "Quantitative data that will be extracted will include publication year, study type (primary, secondary, or grey literature), country of origin, age group, competition level, study design, study description, surgical procedure (if applicable), analysis design, and key findings. Qualitative data will be extracted through qualitative synthesis of related topics."

On page 10 lines 49-54, greater clarity has been added for quantitative data synthesis as follows, "Specifically, quantitative data will be extracted, sorted into relevant themes (e.g. musculoskeletal health, mental health, physiological health), and descriptively reported.⁴⁵

On page 10-11 lines 54-4, greater clarity has been added for qualitative data synthesis as follows, "Qualitative data will be synthesized in excel, through a six stage process. This six stage process includes becoming familiar with the data, generating initial codes, searching for themes, reviewing themes, defining themes, and writing up.^{47 48}

Reviewer comment 2.9: There are up-to-date reviews of sport health and wellbeing missing from the background e.g. Mansfield et al (2018) on sport and dance, and Oja's work on cycling. I do think this kind of review protocol needs to reflect the current literature more specifically.

Oja, P., Titze, S., Bauman, A., De Geus, B., Krenn, P., Reger-Nash, B. and Kohlberger, T., 2011. Health benefits of cycling: a systematic review. *Scandinavian journal of medicine & science in sports*, 21(4), pp.496-509.

Mansfield, L., Kay, T., Meads, C., Grigsby-Duffy, L., Lane, J., John, A., Daykin, N., Dolan, P., Testoni, S., Julier, G. and Payne, A., 2018. Sport and dance interventions for healthy young people (15–24 years) to promote subjective well-being: a systematic review. *BMJ open*, 8(7), p.e020959.

Author response 2.9: The authors would like to thank the reviewer for bringing this to our attention. These two studies have been added to the background.

Author action 2.9: On page 5 lines 31-45, the references have been added as follows, "The link between sport participation and beneficial health outcomes have been synthesised in previous systematic reviews for golf, cycling, and sport and dance.³⁵⁻³⁷ These studies found that participation in these activities had a positive relationship with physical health and wellbeing.³⁵⁻³⁷

Reviewer comment 2.10: The background also needs to make clear what the impact of this evidence can really be for cricket players, organisations and the sport sector.

Author response 2.10: The authors would like to thank the reviewer for this comment. Greater clarity has been added regarding the specific potential impact of this study.

Author action 2.10: On page 5 lines 26-45, greater clarity has been added as follows, “This overview would also enable key stakeholders (including cricket participants, health professionals and sporting bodies) to make evidence informed decisions relating to cricket participation. Specifically, these data will inform stakeholders on the health and wellbeing risks and benefits of cricket participation in order to make individual and organizational decisions on the viability of promoting cricket participation as a health enhancing form of physical activity at different standards-of-play and for different age-groups. Further, identifying the gaps in the literature will allow specific cricket related research to be initiated to improve cricket participant health and wellbeing.”

Reviewer comment 2.11: I can see a cross reference to relevant methods for scoping reviews but it is rather a vague commentary in the text. It’s difficult to know what standards are being followed at every stage. There is information missing such as date ranges for searches for example. Sorry if I have missed this but referring to standards and guidelines rather than papers that have employed scoping reviews will strengthen this protocol.

Author comment 2.11: The authors would like to thank the reviewer for bringing this to our attention. Search dates have been added to the methods. Also, specific scoping review methodological studies have been added to the methods to increase methodological rigor. Specific studies included are Arksey et al. and Levac et al. (see full references below). Greater specificity and clarity in data extraction methodology has been added to the methods as well, with data extraction based off of Munn et al.’s data extraction methodology (see reference below) for quantitative data, and Maguire et al. and Bree et al. for qualitative data.

Arksey H, O'Malley L. Scoping studies: towards a methodological framework. *International journal of social research methodology* 2005;8(1):19-32.

Levac D, Colquhoun H, O'Brien KK. Scoping studies: advancing the methodology. *Implementation science* 2010;5(1):69.

Munn Z, Tufanaru C, Aromataris E. JBI's systematic reviews: data extraction and synthesis. *AJN The American Journal of Nursing* 2014;114(7):49-54.

Maguire M, Delahunt B. Doing a thematic analysis: A practical, step-by-step guide for learning and teaching scholars. *AISHE-J: The All Ireland Journal of Teaching and Learning in Higher Education*. 2017 Oct 31;9(3).

Bree RT, Gallagher G. Using Microsoft Excel to code and thematically analyse qualitative data: a simple, cost-effective approach. *AISHE-J: The All Ireland Journal of Teaching and Learning in Higher Education*. 2016 Jun 30;8(2).

Author action 2.11: Page 6-7 Lines 56-4, the dates for the study have been added to the methods as follows, “The proposed study is planned to be commenced during September 2019, and is estimated to conclude in March 2020..”

Page 6 lines 49-57, specific scoping review methodological studies have been added as references to the methods as follows, “The framework adopted for this scoping review follows existing best practice methodology.38-42 The methodology was guided by the recommended five-stage process: identify the research question; identify relevant studies; select articles using a priori inclusion/exclusion criteria; chart data; collate, summarize, and report results.38 41 42”

Page 10 lines 10-20, greater clarity has been added to the data extraction methodology as follows, “Quantitative data that will be extracted will include publication year, study type (primary, secondary, or grey literature), country of origin, age group, competition level, study design, study description,

surgical procedure (if applicable), analysis design, and key findings. Qualitative data will be extracted through qualitative synthesis of related topics.”

Page 10 lines 49-54, greater clarity has been added for quantitative data synthesis as follows, “Specifically, quantitative data will be extracted, sorted into relevant themes (e.g. musculoskeletal health, mental health, physiological health), and descriptively reported..45

Page 10-11 lines 53-4 greater clarity has been added for qualitative data synthesis as follows, “Qualitative data will be synthesized in excel, through a six stage process. This six stage process includes becoming familiar with the data, generating initial codes, searching for themes, reviewing themes, defining themes, and writing up.47 48

Reviewer comment 2.12: Inclusion / exclusion criteria are probably best represented in a table.

Author response 2.12: The authors would like to thank the reviewer for this comment. The inclusion and exclusion criteria are now presented in a table, as suggested. Please see Table 1 for inclusion and exclusion criteria.

Reviewer comment 2.13: On page 9 it read ‘the scoping review is novel’. Do you mean ‘this scoping review of XXXX is novel’? Needs clarifying because a scoping review is not novel in and of itself.

Author response 2.13: Thank you for this comment. Yes, you are correct in that scoping reviews are not novel; but, investigating cricket health and wellbeing through a scoping review methodology is. This has been clarified.

Author action 2.13: Page 11 line 24 the statement has been revised as follows, “This scoping review of cricket health and wellbeing is novel and will provide an overview of associations between cricket, health and wellbeing.

Reviewer comment 2.14: The funding seems to refer to a grant linked to Arthritis – is this health condition relevant to the rationale for the review.

Author comment 2.14: The authors would like to thank the reviewer for this question. Our specific funding relates to the Centre for Sport, Exercise and Osteoarthritis Research Versus Arthritis. The interplay between sports participation, post-traumatic osteoarthritis, exercise limitations and wellbeing, is of high relevance to the aims of the Centre for Sport, Exercise and Osteoarthritis Research Versus Arthritis. Consequently, this funding is related to the rationale for this work, in addition to a large multi-institution body of work investigating the relationship between sport participation, musculoskeletal health and quality of life including a number of primary research studies across a variety of sports.

Reviewer comment 2.15: I do think much more needs to be said about the way in which this particular topic was developed with the identified stakeholders if it’s relevant. A stronger link to the use of the evidence by stakeholders would then strengthen the case for the scoping review approach.

Author response 2.15: The authors would like to thank the reviewer for this comment. This specific scoping review protocol is based off of previous stakeholder work involving qualitative interviews (see references below). From these initial qualitative studies, we identified a need for further research to better understand the relationship between cricket participation, health and wellbeing. Further, informing research strategy throughout the process with stakeholders allows for improved research pertinence and application. Due to this, a stakeholder group involving former cricket players, coaches, clinicians, many of whom from multiple countries, will be involved in this scoping review. These stakeholders will be involved in assisting in interpreting the results, and helping in disseminating findings. Stakeholder involvement has been added at the beginning of the methods and within the conclusion to further elucidate this process.

Filbay SR, Bishop FL, Peirce N, Jones ME, Arden NK. Physical activity in former elite cricketers and strategies for promoting physical activity after retirement from cricket: a qualitative study. *BMJ Open*. 2017 Nov 17;7(11).

Filbay SR, Bishop FL, Peirce N, Jones ME, Arden NK. Common attributes in retired professional cricketers that may enhance or hinder quality of life after retirement: a qualitative study. *BMJ Open*. 2017 Jul 26;7(7).

Author action 2.15: Page 5-6 lines 59-27, the statement has been added as follows, “
Stakeholder Involvement

Findings from two qualitative studies investigating the relationship of physical activity and quality of life in former elite cricketers,38 39 highlighted a need for further research investigating the relationship between cricket participation and health. It was determined that a scoping review would provide valuable information regarding the relationship between cricket participation, health and wellbeing, whilst identifying key knowledge gaps to guide future research agendas. An international stakeholder group will be established, comprising current and former cricketers, cricket coaches, sports medicine professionals, cricket-related researchers and representatives from cricket sporting bodies. This key stakeholder group will meet virtually, to discuss preliminary results and interpretation of findings, review a draft of the manuscript, and provide input into the plan for dissemination of research findings.”

Page 11-12 lines 57-6, the statement has been added as follows, “The aims and methodological study design were created in concordance with cricket stakeholders (current and former cricketers, physicians, physiotherapists, and governing bodies) in order to have a greater understanding of the relationship between cricket participation, health and wellbeing.”

Reviewer: 2

Reviewer Name: Paul Downward

Institution and Country: School of Sport, Exercise and Health Sciences, Loughborough University, UK.

Please state any competing interests or state ‘None declared’: None declared

Please leave your comments for the authors below

Dear authors,

Reviewer comment 3.1: There are a few minor issues that I feel that you should address in the manuscript. The most important one is to clarify the distinction between a scoping and systematic review. The former as a more preliminary and exploratory investigation (See Rebecca Armstrong, Belinda J. Hall, Jodie Doyle, Elizabeth Waters, ‘Scoping the scope’ of a cochrane review, *Journal of Public Health*, Volume 33, Issue 1, March 2011, Pages 147–150,<https://doi.org/10.1093/pubmed/fdr015>). However, you argue that the scoping review is the best methodology to use in your case. Why? and would not a systematic review be better. Should this work be a precursor to that?

Author response 3.1: The authors would like to thank the reviewer for their due diligence in reviewing this manuscript. Scoping reviews are designed to offer a systematic approach to investigate broad questions and identify gaps in the literature. Further, scoping reviews are designed to create an understanding of the breadth of the literature and direction for future research. This is in comparison to systematic reviews and/or meta-analyses which are designed to amalgamate the literature on a focused study question, where large consideration to study design and quality is given. The aim of this scoping review is to investigate the relationship between cricket participation, health, and wellbeing at all ages and standards of play, and to identify research gaps. Due to the broad nature of these aims and the wealth of literature on this topic, a scoping review is the best systematic

methodology to be employed. Within the abstract, greater clarity on the reasoning behind using a systematic scoping methodology has been added.

Author action 3.1: Page 2 lines 28036, the abstract has been further clarified as follows, “Due to the broad nature of our research question, the large number of health outcomes assessed within the cricket literature, and to facilitate identification of research gaps, a scoping review methodology was utilised.”

Page 6 lines 40-49, the manuscript has been further clarified as follows, “The purpose of a scoping review is to describe all available evidence underpinning a given research question drawing upon research from all possible sources, consequently scoping reviews are broad in nature. This is in comparison to a systematic review which can only investigate one specific topic; as a result, a scoping review methodology was commenced.^{38 39}”

Reviewer comment 3.2: The impact of cricket on health is likely to be highly geographically specific, reflecting its historic development. Is allowance made for this both in terms of the language of research papers extracted and the context in which cricket is played in terms of the social determinants of health.

Author response 3.2: The authors would like to thank the reviewer for this comment. One of the data points that will be extracted is geographic origin of each study. Geographic origin will then be tabulated and reported within the manuscript. Further clarification on geographic origin extraction has been added to the manuscript. In terms of language, due to the authors only speaking English, only studies written in English will be included in the review.

Author action 3.2: Page 10 lines 10-19, “Quantitative data that will be extracted will include publication year, study type (primary, secondary, or grey literature), country of origin, age group, competition level, study design, study description, surgical procedure (if applicable), analysis design, and key findings.”

Page 10-11 lines 49-4, greater clarity has been added for quantitative data synthesis as follows, “Specifically, quantitative data will be extracted, sorted into relevant themes (e.g. musculoskeletal health, mental health, physiological health), and descriptively reported. 45

Reviewer comment 3.3: More clarification is need on the links between the preliminary and actual searches discussed on p.6.

Author response 3.3: The authors would like to thank the reviewer for this suggestion. Further clarification has been added to link the preliminary search and the final search strategy. Further, an Appendix has been added to better describe the full search strategy.

Author action 3.3: Page 7-8 lines 59-10, the statement has been revised as follows, “The final search strategy was created to keep the search broad for greatest inclusion, while excluding specific irrelevant studies identified through the preliminary search. A medical librarian was consulted on specific search syntax; however, the final search was created by the authors.”

Reviewer comment 3.4: The nature and role of the librarian and their access to research papers should be clarified on p.8 line 36.

Author response 3.4: The authors would like to thank the reviewer for this comment. A medical librarian assisted in ensuring the search syntax was appropriate for each database. . In terms of research paper access, papers are available through the University’s online portal access. If papers are not available through this specific portal, library access is also available through collaborating institutions in Sweden, Australia, and the United States, allowing for greater reach. If the papers are still not available through these avenues, the authors will be contacted to request full-text, and if required, inter library loan will be used to gain access to the papers. This information has been further clarified within the methods.

Author action 3.4: Page 9 lines 36-51, the statement has been revised as follows “Firstly, we will attempt to access articles through university online library portals. The online library portals will be available through collaborating institutions in the United Kingdom, Sweden, Australia, and the United States. If the article cannot be retrieved through the university online library portals, the authors will be contacted to request full-text, and if required inter library loan with the assistance of a librarian will be attempted. If a full-text article cannot be retrieved following consultation with a librarian, it will be excluded from the review.⁴⁰”

Page 8 lines 6-11, the librarian’s role was further clarified as follows, “A medical librarian assisted by ensuring the search syntax was appropriate for each database.”

Reviewer comment 3.5: There are some minor issues of expression to resolve.

p.2 line 19 'The Methodology...'

Author response 3.5: The authors would like to thank the reviewer for this comment. The sentence has been revised per your editorial comment.

Author action 3.5: Page 2 lines 33-38, the sentence has been revised as follows, “The methodology of this paper was informed by previous scoping review protocols and best practice methodological frameworks.”

Reviewer comment 3.6: p.2 line 38 'data being available' not 'data is available...'

Author response 3.6: Thank you for this comment, the statement has been revised as per your suggestion.

Author response 3.6: Page 7 lines 3-7, the sentence has been revised as follows, “This research is exempt from ethical approval since it is a review of previously published articles.”

Reviewer: 3

Reviewer Name: Ivan Cavero-Redondo

Institution and Country: Universidad de Castilla-La Mancha

Please state any competing interests or state 'None declared': None

Please leave your comments for the authors below

Reviewer comment 4.1: I think that the topic is of little scientific interest and it also seems that it is not necessary to develop a protocol for a scoping review for this topic. I think it would have something more of interest if it were a systematic review of studies whose objective was to analyze the effect of cricket in the wellbeing of its participants or any particular health outcome.

Author response 4.1: The authors would like to thank the reviewer for taking the time to review this manuscript. The authors believe that in order to increase the transparency and scientific rigor of this study, a protocol was needed. Notably, scoping reviews are not included in publicly available review databases. Consequently, a full published protocol in a peer-reviewed journal was necessary.

In response to using a scoping review methodology, scoping reviews are designed to offer a systematic approach to investigate broad questions and identify gaps in the literature. Further, scoping reviews are designed to create an understanding of the breadth of the literature and direction for future research. This is in comparison to systematic reviews and/or meta-analyses which are designed to amalgamate the literature on a focused study question, where large consideration to study design and quality is given. The aim of this scoping review is to investigate the relationship between cricket participation, health, and wellbeing at all ages and standards of play, and to identify research gaps. Due to the broad nature of these aims and the wealth of literature on this topic, a

scoping review is the best systematic methodology to be employed. Within the abstract, greater clarity on the reasoning behind using a systematic scoping methodology has been added.

VERSION 2 – REVIEW

REVIEWER	Louise Mansfield Brunel University London I hold and compete for funding for research on community sport, health and wellbeing.
REVIEW RETURNED	20-Sep-2019
GENERAL COMMENTS	The authors have addressed my review comments in detail.